# Nutritional Regulation of Human Brown Adipose Tissue

**DOI:** 10.3390/nu13061748

**Published:** 2021-05-21

**Authors:** Karla J. Suchacki, Roland H. Stimson

**Affiliations:** University/BHF Centre for Cardiovascular Science, The Queen’s Medical Research Institute, University of Edinburgh, 47 Little France Crescent, Edinburgh EH16 4TJ, UK; roland.stimson@ed.ac.uk

**Keywords:** brown adipose tissue, obesity, type 2 diabetes, dietary (calorie) restriction, nutrition, energy expenditure

## Abstract

The recent identification of brown adipose tissue in adult humans offers a new strategy to increase energy expenditure to treat obesity and associated metabolic disease. While white adipose tissue (WAT) is primarily for energy storage, brown adipose tissue (BAT) is a thermogenic organ that increases energy expenditure to generate heat. BAT is activated upon cold exposure and improves insulin sensitivity and lipid clearance, highlighting its beneficial role in metabolic health in humans. This review provides an overview of BAT physiology in conditions of overnutrition (obesity and associated metabolic disease), undernutrition and in conditions of altered fat distribution such as lipodystrophy. We review the impact of exercise, dietary macronutrients and bioactive compounds on BAT activity. Finally, we discuss the therapeutic potential of dietary manipulations or supplementation to increase energy expenditure and BAT thermogenesis. We conclude that chronic nutritional interventions may represent a useful nonpharmacological means to enhance BAT mass and activity to aid weight loss and/or improve metabolic health.

## 1. Introduction

Obesity and its associated comorbidities such as type 2 diabetes (T2DM), dyslipidemia and hypertension have a substantial adverse impact on health outcomes. In the UK, obesity alone accounts for >30,000 deaths per year and decreases life expectancy by nine years [1,2]. Obesity occurs when energy intake chronically exceeds energy expenditure (EE), resulting in the accumulation of excessive white adipose tissue (WAT). Despite numerous public health initiatives, rates of obesity continue to rise and currently one quarter of the adult UK population are obese and approximately two-thirds are classed as at least overweight [1]. Current anti-obesity medications act through reducing energy intake, either by inhibiting appetite or dietary fat absorption, but only two agents are licensed in the UK (orlistat and naltrexone/bupropion) and novel approaches are urgently required [3]. However, the recent discovery of brown adipose tissue (BAT) in adult humans [4,5,6] offers an exciting new strategy to treat obesity and associated metabolic disease by increasing EE (Figure 1). BAT is a thermogenic and highly plastic tissue and understanding the factors that control expansion of BAT mass and activation are key to determining the therapeutic potential of BAT. In this review we will focus on the nutritional regulation and dysregulation of BAT activity in health and in metabolic disease in humans.

## 2. White and Brown Adipose Tissue

Adipose tissue exists in distinct subtypes typically classified as WAT or BAT (Figure 1). WAT is the primary organ in the body responsible for energy storage but WAT is also an important endocrine organ which regulates biological processes such as metabolic homeostasis and immunity [12]. White adipocytes contain a single large lipid droplet and few mitochondria. In contrast, BAT is a specialized thermogenic organ that increases EE to generate heat, maintaining body temperature in a cold environment [13]. In line with their thermogenic role, brown adipocytes contain substantially greater numbers of mitochondria and numerous small lipid droplets. Cold exposure stimulates BAT thermogenesis through the sympathetic nervous system via β-adrenergic receptors (classically β3-AR but also β1- and β2-AR) [14,15,16]. This subsequent signaling cascade leads to activation of the unique thermogenic protein uncoupling protein 1 (UCP1) which is situated in the inner mitochondrial membrane. UCP1 uncouples mitochondrial respiration from adenosine triphosphate (ATP) production, allowing the mitochondrial membrane potential to be transduced to heat. BAT thermogenesis requires a high metabolic demand, as such BAT utilizes a number of metabolic substrates including intracellular triglycerides, circulating free fatty acids (FFA) and glucose [17].

In addition to white and brown adipocytes, thermogenic ‘beige’ adipocytes can form under suitable stimulation such as cold exposure, for example in suprascapular, anterior subcutaneous and inguinal WAT depots in rodents in a process termed ‘browning’ (Figure 1b) [18]. These inducible beige adipocytes share similarities with ‘classical’ brown adipocytes such as containing multiple lipid droplets. Importantly, beige adipocytes also demonstrate increased glucose and FFA uptake in response to cold and contain UCP1 positive mitochondria, albeit to a lesser extent than brown adipocytes [19].

In rodents, BAT is primarily located in the interscapular region (Figure 1b) and has a critical role in nutritional homeostasis [20,21]. BAT is an essential organ to maintain body temperature in rodents due to their inability to defend body temperature, in part due to their greater body surface-to-weight ratio than larger mammals such as humans [22]. In addition, the important role of BAT in energy balance in rodents was identified >40 years ago [23,24]. Subsequent studies reported dysregulated BAT function in other models of obesity [25], while genetic disruption of UCP1 induced obesity in mice at thermoneutrality [20] as did ablation of BAT [21]. These studies, amongst others, underscore the importance of BAT in the maintenance of energy balance and metabolic health.

Interscapular BAT is also present in human infants (Figure 1b) which regresses soon after birth [10,26]. BAT depots are localized in the supraclavicular, axillary and peri-adrenal regions in the infant and smaller depots of BAT have also been recorded behind the sternum and along the spine in human newborns [7,8,10]. Until two decades ago, BAT was not thought to be present in adult humans; however, the development of positron emission tomography/computed tomography (PET/CT) using the metabolic tracer ^18^F-fluorodeoxyglucose (^18^F-FDG) to diagnose malignancies led to the incidental identification of BAT in a proportion of patients [27,28]. Human BAT is located primarily in the cervical, supraclavicular, paravertebral, periaortic and perirenal depots (Figure 1c) [29,30]. Morphologically, adult human BAT is a heterogeneous blend of multilocular and unilocular adipocytes [31]. Functionally, human BAT is also activated by cold exposure [4,5,6,32] and increases EE, insulin sensitivity and lipid clearance, highlighting the positive role of BAT in metabolic health [33,34,35]. Whilst the regulation of human BAT demonstrates certain similarities to rodents, for example both are activated by cold exposure [32] and are under sympathetic control [4], there are also species-specific differences [36], emphasizing the need to study BAT in humans to fully understand the mechanisms regulating this tissue. At present, the mechanisms controlling human BAT expansion and activation are not fully understood. One of the reasons behind our limited understanding of human BAT is the difficulty measuring in vivo activity. The majority of studies have used ^18^F-FDG PET/CT to measure BAT activity, either by analysis of retrospective clinical imaging datasets or in dedicated cooling studies [4,37,38]. Dedicated cooling studies use a variety of methods to activate BAT, such as air (cold room)- or water-cooling protocols (water-perfused suit/blanket). Retrospective studies have a caveat, in that scans are performed at room temperature, when BAT is not as active although it still demonstrates greater glucose and FFA uptake than WAT [39,40]. In addition, the use of other novel PET tracers such as ^18^F-fluoro-6-thia-heptadecanoic acid (^18^F-FTHA) and ^11^C-acetate have demonstrated important biological insights such as FFA uptake and oxidative metabolism [35]. Due to certain limitations with PET/CT, alternative modalities have been sought to assess BAT morphometry and function, in part to reduce radiation exposure. To date, ^18^F-FDG PET magnetic resonance imaging (MRI) [41], MRI [42], MR spectroscopy [43], infrared thermography [44], multispectral optoacoustic tomography [45] and microdialysis [39] have been developed as alternative/complementary techniques, some of which have provided additional insights into BAT physiology.

## 3. Dysregulation of BAT by Altered Nutritional Status

### 3.1. Overnutrition, Obesity and T2DM

The uptake of ^18^F-FDG into BAT is substantially reduced with aging, obesity and T2DM [5,32,46,47]. Uptake of ^18^F-FTHA is reduced in obese subjects at both room temperature and during acute cold exposure (achieved using blankets perfused with water cooled to ~10 °C for 2 h), in keeping with reduced BAT mass and activity [40]. In humans, fat accumulation and distribution in obesity can be divided into those with central obesity (apple-shape), whereby subjects have a high waist-to-hip ratio (WHR), and peripheral obesity (pear-shaped, with a lower WHR) [48]; those with central obesity have a greater cardiometabolic risk [49,50]. Subjects with central obesity also have decreased ^18^F-FDG uptake by BAT at room temperature [51] and UCP1 expression in human BAT negatively correlates with greater central/peripheral fat distribution [52]. These data suggest an association between BAT activity and WAT distribution, however, given that these data are correlative, it is not currently possible to determine if BAT regulation is driving altered WAT distribution or vice versa. BAT has also been implicated in altered BAT function in the increased cardiometabolic risk of central obesity (Figure 2). Cold-induced BAT activation in mice lowers triglyceride and cholesterol levels https://www.nature.com/articles/s41591-020-1126-7-ref-CR1 (accessed on 21 April 2021), protecting mice from the development of atherosclerosis [53,54]. In humans, high BAT activity (^18^F-FDG uptake during cold exposure) correlates with reduced cardiovascular risk factors and subsequent development of early markers of atherosclerosis [55], while a large retrospective study (>50,000 PET CT scans) showed that the presence of BAT was associated with a lower prevalence of cardiometabolic disease particularly in obese subjects [56]. However, it remains unclear how BAT is mechanistically connected to cardiometabolic risk. Two possibilities are that (1) BAT may communicate with other organs to reduce metabolic risk via the secretion of batokines [57], and (2) BAT removes circulating glucose and lipids to improve cardiometabolic health. Alternatively, it is possible that genetic factors that are associated with BAT development and activity may also be associated with cardiometabolic disease. Further research is therefore required to determine any causative role of BAT in protecting against cardiometabolic disease.

While reduced BAT glucose uptake may be a measure of decreased thermogenesis, BAT is an insulin sensitive organ and early studies in rabbits showed that insulin infusion had a direct effect on glucose uptake by BAT during mild cold exposure (20 °C for 48 h) but had little effect on glucose uptake in warm conditions [58]. In humans, insulin enhanced ^18^F-FDG glucose uptake into BAT approximately five-fold during warm conditions (albeit to a lesser extent than during cold exposure at 17 °C for 2 h), although whether insulin activates human BAT thermogenesis is unclear [59]. ^18^F-FDG uptake by BAT correlates with whole-body insulin sensitivity so BAT glucose uptake may reflect insulin resistance [60]. In T2DM subjects, despite reduced ^18^F-FDG uptake by BAT these subjects had similar total EE and BAT oxidative metabolism compared to control subjects [47]. However, UCP1 expression in human BAT negatively correlates with glycated hemoglobin (HbA1c) [52]. Overall, these data suggest that whilst it is likely that BAT activity/volume is reduced in obesity and T2DM, importantly BAT retains functionality. In addition, repeated cold exposure (~15 °C for 10 days; 2 h on day 1, 4 h on day 2 and 6 h on days 3–10) increases ^18^F-FDG uptake by BAT in subjects with obesity and T2DM, suggesting either that BAT mass can be recruited or that dormant BAT can become functional [61,62]. This may represent a treatment to increase BAT activity and EE, promote weight loss and improve metabolic health.

The distribution and volume of WAT contributes to ethnic differences in T2DM and cardiovascular disease [63,64,65,66], whereby the risk of T2DM is increased in Asian and black ethnic groups compared to Caucasians [67]. Differences in BAT volume and activity have been observed in different ethnic groups which could increase their susceptibility to obesity, T2DM and/or cardiovascular disease. For example, lean male Dutch national south Asians had ~25% lower energy expenditure during warm conditions than age- and BMI-matched Caucasian participants, decreased BAT volumes (as measured by ^18^F-FDG PET/CT during 2h cold exposure) and increased shivering thermogenesis [68]. Genetic studies have identified *UCP1* variants associated with nonshivering thermogenesis, while the *UCP1* haplotype with the highest thermogenic efficiency was more commonly found in populations from colder habitats [69]. These data support the hypothesis of evolutionary regulation of BAT, where descendants from countries with the absence of extreme cold have reduced BAT volume and activity. Consequently, while individuals with south Asian or African heritage have an increased prevalence of obesity and T2DM it is possible that strategies to activate BAT would be less successful in these populations.

### 3.2. Weight Loss and Caloric Restriction

Similar to the effect of repeated cold exposure in obese subjects, weight loss (~29% weight loss following one year) following bariatric surgery increases ^18^F-FDG uptake by BAT [70], in keeping with recruitment of BAT. A more common, less invasive, method for weight loss is dietary calorie restriction (CR). CR is defined as the reduction in energy intake below normal ad libitum levels without malnutrition or deprivation of essential nutrients. Rodent, primate and human studies have suggested that CR may delay the onset of age-related cardiovascular and neurodegenerative diseases and can induce the remission of T2DM [71,72,73]. The mechanisms through which CR decreases the risk of chronic disease in humans remains incompletely understood, primarily due to the lack of long-term studies [74]. In mice, 12–20 weeks of CR decreases metabolic rate, BAT mass and core body temperature [75,76,77], however, in rats, 40% CR for ~6 and 26 months caused BAT hypertrophy without altering *Ucp1* gene expression [78]. Changes in BAT mass do not necessarily translate to BAT activity, and it is unclear if CR in lean animals has a beneficial effect on BAT activity. CR in mice induces the development of beige fat in subcutaneous and visceral adipose tissue which may be required to increase body temperature during feeding [79]. In humans, a two-phase dietary intervention (eight-week very low-calorie diet ~780–1000 Kcal/day and six-month weight maintenance) decreased thermogenic markers in abdominal WAT, although these markers were not measured in BAT [80]. Due to the limited data available the effects of CR on BAT activity in normal weight individuals is unknown, but in obese subjects, we speculate that CR may lead to the recruitment and activation of BAT, counteracting the reduced BAT mass observed in obesity.

### 3.3. Undernutrition and Lipodystrophy

There is also evidence that BAT activity is dysregulated in underweight individuals. Anorexia nervosa (AN) is an eating disorder of chronic starvation causing severe depletion of body fat and fat-free mass (Figure 2). Women with AN (either current or recovered) may have a lower prevalence of ^18^F-FDG uptake by BAT during cold exposure (using a cooling vest at 17 °C for 2 h) compared with normal weight controls, in keeping with reduced BAT activity although the sample size was small [81]. Defects in skeletal muscle thermogenesis have also been identified in AN patients [82]. A retrospective analysis of over 15,000 ^18^F-FDG PET/CT cases identified reduced BAT activation in underweight patients (BMI <16 kg/m^2^), while those with a BMI between 18–21 kg/m^2^ had the greatest ^18^F-FDG uptake [83]. Young patients with constitutional leanness (mean BMI ~16 kg/m^2^), but not AN patients, demonstrate ^18^F-FDG uptake by BAT [84], potentially suggesting that AN reduces BAT activity rather than a low BMI per se. However, due to the lack of histological analysis it is possible than AN subjects do have BAT, albeit inactive.

Another group of disorders with substantial reductions in fat mass are the lipodystrophy syndromes, these patients may have fat loss throughout the entire body (congenital generalized lipodystrophy; CGL) or in discrete regions (familial partial lipodystrophy; FPLD) [85]. Lipodystrophy syndromes are associated with insulin resistance, diabetes, dyslipidemia, kidney disease and nonalcoholic fatty liver disease and there is also evidence of dysregulated BAT activity (Figure 2). CGL type 2 (CGL2) is caused by mutations in the Berardinelli–Seip congenital lipodystrophy 2 gene (BSCL2) [86]. BSCL2 is more highly expressed in murine BAT than other adipose depots, highlighting a potentially important role in thermogenesis [87]. BSCL2 is essential for acute and chronic cold-induced BAT activation in mice and prevents BAT necroptosis [88], it is unknown though whether BAT activity is altered in patients with CGL2. However, BAT activity may be decreased in patients with type 2 familial partial lipodystrophy (FPLD2), whereby cold exposure (2 h intermittent feet immersion, 5 min in/5 min out in cold water at ~4 °C) did not result in ^18^F-FDG uptake by BAT and differentiated pre-adipocytes obtained from the supraclavicular BAT depots had lower UCP1 levels than control subjects [89]. These data suggest that BAT activity is reduced in lipodystrophy patients and that specific mutations that result in lipodystrophies have direct effects on brown adipocyte function. Further investigation into diseases of adipose distribution in the human may also provide unique insights as to how specific WAT depots influence BAT mass and thermogenesis.

## 4. Dietary Components That Alter BAT Thermogenesis

While overnutrition and undernutrition regulate BAT activity, dietary composition also has an important effect on BAT function. Understanding how dietary components alter BAT mass and activation may identify targets for therapeutic manipulation without the need to develop pharmacological agents [90].

### 4.1. Diet-Induced Thermogenesis

The increase in EE above the basal metabolic rate in response to food intake is referred to as diet-induced thermogenesis (DIT). Thermogenesis following a meal is divided into (1) obligatory thermogenesis and (2) facultative thermogenesis. Obligatory thermogenesis is the necessary accompaniment of all metabolic processes including digestion, absorption and storage of ingested nutrients; whereas facultative thermogenesis is regulated by hypothalamic centers and is the increase in EE in response to dietary intake which varies with food composition and volume of food, which we refer to as DIT [91]. In rodents, DIT has been suggested for four decades, whereby a cafeteria diet increased EE and interscapular skin temperature [23] and a single low-protein high-carbohydrate meal increased BAT respiration [92]. However, other studies found no increase in BAT thermogenesis in rats fed a cafeteria diet [93] or a low-carbohydrate, high-fat diet (HFD) [94]. DIT in adult humans accounts for 5–15% of daily EE [95] and recent work has identified that BAT contributes to DIT, as evidenced by increased BAT oxygen consumption and blood flow following a mixed high-carbohydrate meal. In fact, BAT activity was increased to the same extent as seen during a cold stress (2 h, individualized cooling protocol) [96]. In addition, BAT-positive individuals have higher DIT compared with BAT-negative subjects [97]. The acute effects of DIT observed in both animals and humans in response to a meal composed of varying macronutrient content is likely to differ when administered chronically. In obese children (mean BMI ~26 kg/m^2^, aged ~10 years) a high fat meal (48% fat) resulted in lower thermogenesis (measured by indirect calorimetry) compared to an isocaloric and isoproteic low-fat meal [98]. Furthermore, rodents maintained on a HFD have decreased BAT energy consumption and accumulate WAT [99,100].

### 4.2. Macronutrient Content of Meals

Some of the conflicting data from rodent studies investigating DIT may be due to the specific macronutrient content of the meals as this may alter post-prandial BAT thermogenesis. Here we will provide an overview of the effects of individual macronutrients and bioactive compounds on BAT function.

#### 4.2.1. Carbohydrate

In rats, a low-protein, high-carbohydrate diet (74% carbohydrate, 6% protein) for 15 days increased basal and noradrenaline-stimulated thermogenesis and increased *Ucp1* gene expression in BAT by 60% compared to rats fed an isocaloric diet containing 63% carbohydrate and 17% protein, suggesting increased BAT activity [101]. There are no studies investigating the effect of dose-dependent carbohydrate diets on BAT function in humans, however, indirect evidence suggests that carbohydrate likely increases BAT thermogenesis. Two independent studies demonstrated that a single carbohydrate rich meal (78% ~ 1622 kcal and 58% ~ 542 kcal carbohydrate, respectively) stimulated BAT ^18^F-FDG uptake at room temperature [96,102] and increased BAT oxidative metabolism [96]. In addition, carbohydrate consumption stimulates insulin secretion and insulin increases glucose uptake by BAT [59]. Insulin has direct effects on the sympathetic nervous system and increases BAT activity and thermogenesis [103,104]. It is therefore possible that dietary carbohydrate may acutely increase BAT activity in part via insulin secretion.

#### 4.2.2. Fat

In rodents, HFDs of different fat content and duration increase *Ucp1* mRNA and protein expression in BAT [105]. However, not all dietary fats elicit identical responses in BAT. In rats, supplementation of different C18 fatty acids for seven days failed to increase *Ucp1* mRNA expression in BAT despite inducing weight loss, although 2-hydroxyoleic acid increased *Ucp1* mRNA expression in WAT [106], potentially in keeping with browning of WAT. Fat-induced activation of BAT thermogenesis may be mediated by cholecystokininactivation (CCK) which is a satiety hormone secreted by the I-cells of the upper small intestine following the consumption of both fat and protein [107]. CCK activates sympathetic nerves in BAT, increasing BAT thermogenesis in rats [108,109] through activation of the CCKA receptor [110].

Few studies in humans have investigated the effect of dietary fat on adipose thermogenesis. Eicosapentaenoic acid (an omega-3 fatty acid) increases UCP1 expression in white abdominal and mammary differentiated pre-adipocytes obtained from lean women, but this effect was not investigated in brown adipocytes [111]. Human BAT does utilize dietary fatty acids (DFA) as demonstrated by oral ^18^F-FTHA administration during 4 h cold exposure in (using a water-cooled suit at ~18.0 °C) [112]. However, ^18^FTHA uptake by BAT was not increased following a four-week cold acclimation protocol and BAT only accounts for a small proportion (<1%) of total body DFA clearance [112] and whether these DFAs stimulate thermogenesis at room temperature is unknown.

#### 4.2.3. Protein

In rodents a single low-protein high-carbohydrate meal increases [^3^H] norepinephrine ([^3^H]-NE) turnover in BAT [113] and long-term high protein diets (8–16 weeks; 51–70% protein) increase BAT *Ucp1* gene expression and decrease body weight compared to diets with lower protein/carbohydrate ratios [114,115,116,117,118]. However, the specific dietary protein may also play a crucial role in the regulation of BAT function [119]. In humans, increasing dietary protein whilst maintaining a constant carbohydrate intake reduces ad libitum caloric intake [120]. High protein diets are thought to alter energy balance through DIT related satiety [121], aiding weight loss by reducing energy efficiency through increased thermogenesis [95]. However, to date no one has explored the effect of protein on BAT activity and the effect of high protein diets on weight loss in humans is not conclusive [120,122].

### 4.3. Dietary Bioactive Compounds

#### 4.3.1. Capsaicin and Capsinoids

One of the most extensively studied dietary components involved in BAT activation are capsaicin and capsinoids which are found in chili peppers and are potent activators of the capsaicin receptor (transient receptor potential vanilloid 1; TRPV1) [123]. TPRV1 is involved in body temperature regulation in rodents, primates and humans [124,125], and TRPV1 activation has been shown to both increase [126] and decrease [127] BAT thermogenesis. In rats, intravenous injection of the TRPV1 agonist dihydrocapsaicin reduced BAT sympathetic nerve activity, thermogenesis and core body temperature [128]. Contrary to these findings, capsaicin stimulated TRPV1 activation in WAT and induced inguinal beiging in mice [129]. Capsinoids represent a more appropriate supplement in humans due to their nonpungent nature. In mice, capsinoid administration increases BAT temperature, whole body oxygen consumption and fat oxidation, mediated through *Trpv1* [130]. In humans, during cold exposure (19 °C for 2 h, feet placed on an ice block), a single dose of oral capsinoids (9 mg) increased ^18^F-FDG uptake by BAT without altering supraclavicular skin temperature [131]. It remains unclear if capsinoid supplementation alone would be an effective strategy to activate/recruit BAT in adult humans. In one study, a single dose of capsinoids (12 mg) only increased EE in BAT-positive subjects during acute cold exposure (~14.5 °C for 2 h) [132]. However, six weeks of daily capsinoids (9 mg) ingestion increased cold induced thermogenesis (CIT, 19 °C for 2 h) in healthy subjects with low BAT activity, suggesting that long-term capsinoid ingestion may increase BAT mass even in subjects with minimal BAT [133]. A combination of capsinoid supplementation and mild cold exposure may be an effective strategy for recruitment and activation of BAT in humans, but it remains unclear whether capsinoids will induce improvements in other metabolic parameters such as insulin sensitivity as observed during cold exposure.

#### 4.3.2. Tea, Caffeine and Catechins

Catechins are natural phenols with antioxidant activity [134]. A meta-analysis determined that daily catechin-caffeine tea ingestion in humans increased daily EE (by 5%) and whole-body fat-oxidation [135]. In humans, a single acute oral catechin/caffeine beverage (615 mg catechin and 77 mg caffeine) increased whole body EE at a thermoneutral temperature (27 °C) only in BAT-positive subjects, assessed by ^18^F-FDG PET/CT, compared to placebo (0 mg catechin and 81 mg caffeine), suggesting that catechin activates BAT in humans [136]. Furthermore, once daily ingestion of this catechin beverage for five weeks increased CIT (at 19 °C for 2 h, intermittently feet placed on an ice block) in individuals with minimal BAT, suggesting either that daily ingestion of catechin may recruit BAT or enhance CIT in other tissues such as skeletal muscle [136]. Administration of catechin (540 mg/day) for 12 weeks in a double-blind study in women increased BAT density by nearly 20% measured by near-infrared time-resolved spectroscopy (a marker of BAT activity) [137,138]. It will be important to investigate the effect of long-term daily catechin-caffeine tea supplementation on body weight and metabolic parameters to determine the utility of this approach.

#### 4.3.3. Menthol

Menthol is a compound found in mint and is an activator of the calcium-permeable cation channel, transient receptor potential melastatin 8 (TRPM8) [139,140]. TRMP8 is expressed in sensory nerves in the skin, senses cold temperature and functions as a ligand-gated channel to menthol, inducing a cold sensation [141]. TRPM8 is expressed in murine BAT and activation by menthol increases BAT *Ucp1* mRNA and protein expression [142]. In vitro, menthol increased UCP1 expression in human white adipocytes [143] and *in vivo* transdermal menthol administration increased total EE over a 7 h period by indirect calorimetry [144]. Further studies in humans are required to assess if menthol increases BAT activity or whether the activation of TRPM8 via menthol or alternative agonists [145] offers therapeutic potential in the treatment of obesity.

#### 4.3.4. Other Dietary Compounds

Many other dietary compounds have been identified in rodent studies to influence BAT thermogenesis and WAT browning, these include conjugated linoleic acid [146], casein protein [119], curcumin [147,148], garlic powder [149], procyanidin-rich extracts from black soybean seed [150], resveratrol [151] and extracts from ginger family plants [152,153] amongst others [154,155] (reviewed in [156]). While these data are from rodent studies, human studies have also identified dietary compounds with thermogenic potential (Figure 3). Amongst these dietary compounds is cinnamaldehyde (found in cinnamon), which increases EE and postprandial fat oxidation in vivo [157] and increases UCP1 expression in human white adipocytes [158]. Given that oral cinnamaldehyde is well tolerated, cinnamon consumption may be a simple yet effective way to activate thermogenesis. In addition to cinnamaldehyde, acute oral administration of 2.5 mg/kg ephedrine (the active component of the herb Ephedra, a direct and indirect sympathomimetic amine) to lean and obese subjects increased ^18^F-FDG uptake by BAT at room temperature [159]. However, it is unlikely that the ephedrine could be used in the clinical setting due to off target effects such as tachycardia and arrhythmias.

#### 4.3.5. Safety of Dietary Approaches to Activate BAT

Macronutrient and bioactive compound supplementation approaches to increase BAT volume could be effective, safe and widely available, and may represent a more holistic clinical strategy for long-term weight loss/maintenance and improved cardiometabolic health. Chronic alteration of macronutrient content is a more complicated strategy and the safety and long-term effects of high fat, protein or carbohydrate diets in humans is not yet fully understood, particularly as the quality of macronutrients is of crucial importance [160]. Strategies may have to differ in certain patient groups such as those with T2DM, however, high-protein diets (~30%) and ketogenic diets have beneficial effects on weight loss and lipid metabolism which is of benefit to T2DM patients [161,162,163]. The long-term safety of the bioactive compounds discussed in this review have not yet been studied in humans, this will be crucial in order to support therapeutic administration.

Briefly, while capsaicin is a potent irritant and can cause pain, sweating and chest pain [123,164,165], capsinoids are well tolerated in healthy humans, eliciting no change in blood pressure or heart rate [166,167]. Bioactive compounds such as tea catechins have been associated with hepatotoxicity [168], therefore their long-term safety must be investigated [169]. Small quantities of cinnamon have been used to flavor food for centuries with no reports of side effects. The suggested daily intake of the active ingredient of cinnamon, cinnamaldehyde is 1.25 mg/kg and the acute and long-term toxicity is low at nutritional doses [170]. Non-nutritional doses of cinnamaldehyde may cause skin irritation [171] and hepatotoxicity [172].

Therefore, the potential beneficial and detrimental effects of long-term dietary manipulation and supplementation must be studied in patients where increasing BAT mass and activity is of potential therapeutic benefit (e.g. those with obesity and T2DM).

## 5. The Effect of Exercise on BAT Activity

Physical activity is another important component of EE with beneficial effects, as exercise improves cardiovascular health and offers protection against obesity, T2DM and metabolic diseases by the improvement of insulin sensitivity, increased WAT mitochondrial activity, glucose tolerance and the reduction in circulating lipids [173].

Several studies have examined the interplay between exercise and BAT thermogenesis. Activation of BAT and exercise both raise core body temperature [174], while exercise also induces sympathetic activation but with the purpose of increasing cardiac output and blood flow to active skeletal muscles. Exercise can both increase [175,176] and decrease [177] BAT thermogenesis in rodents (reviewed in [178]). It remains unclear why animal studies show contradictory results, however, rodent exercise studies have used different training regimes (both type and duration) to investigate the effect of exercise on BAT function.

In humans, exercise more consistently decreases BAT activity. Endurance athletes have reduced cold-induced ^18^F-FDG uptake by BAT compared to sedentary controls, without induction of UCP1 expression in subcutaneous abdominal adipose tissue, also suggesting a lack of adipose tissue browning [179]. Short term high- and moderate-intensity interval training also decreased insulin-stimulated ^18^F-FDG uptake by BAT [180], but exercise increases glucose uptake by skeletal muscle [181]. It is possible that exercise increases skeletal muscle thermogenesis, reducing the need for BAT to maintain body temperature in humans with resultant regression of BAT mass, however, this is yet to be determined.

There has also been recent interest in the secretion of myokines that can increase adipose thermogenesis. In rodents, exercise results in the beiging of WAT, resulting in increased multilocular lipid droplets and upregulation of thermogenic genes (e.g., *Ucp1, Prdm16*) [182,183] in response to the secretion of myokines such as irisin and meteorin-like [184,185]. However, in humans, exercise does not increase plasma irisin or induce browning in normoglycemic, pre-diabetic, sedentary or endurance trained subjects [179,186]. These data suggest that, at least in humans, exercise-induced release of factors from skeletal muscle does not enhance BAT thermogenesis. However, BAT has a role regulating skeletal adaptations, for example exercise induces release of 12,13-diHOME by BAT in both rodents and humans and activates fatty acid uptake and oxidation in skeletal muscle [187]. Further work is required to determine the interplay between BAT and skeletal muscle in the regulation of thermogenesis.

## 6. Perspective

From the pioneering recognition of BAT in nutritional energetics in the late 1970’s [23,24] to the identification of functional UCP1 positive BAT in adult humans over a decade ago [4,5,6,32,188], our understanding of BAT has evolved immeasurably. In vitro and animal studies have made extraordinary contributions to unravelling the physiology of BAT in conditions of weight loss and gain and the effect of dietary composition and individual dietary components on BAT activity. Rodent studies using ^18^F-FDG and other PET tracers have yet to be employed to directly measure nutritional uptake by BAT and BAT thermogenesis in response to varying macronutrients or individual bioactive compounds. These studies will be helpful to guide dedicated human studies in both males and females [189] to determine the best candidates with thermogenic potential.

To date, evidence from dietary studies in mice and humans have produced some conflicting results and it remains unclear whether dietary components and bioactive compounds have direct effects on BAT thermogenesis. Human BAT shows dysregulation in altered nutritional states such as during caloric excess (obesity) or caloric deficiency (anorexia). Furthermore, there is direct nutritional regulation of BAT activity from both macronutrients, dietary components and bioactive compounds, some of which alter EE. It remains to be determined if the nutritional activation of BAT may have therapeutic potential over chronic administration, or if such treatments would result in weight loss or improve metabolic health. Studies to investigate the effect of varying the macronutrient composition in humans are required in order to determine the role of BAT in DIT but are complex to undertake. It will also be important to undertake dedicated human studies in both males and females and in those of different ethnic origins to determine the candidates with the widest thermogenic potential.

Despite these challenges, nutritional interventions may represent a useful non-pharmacological means to enhance BAT mass and activity and warrant extended investigation. Further research will determine if individual or a combination of compounds could be used as a safe clinical strategy for long-term weight loss/maintenance and improved metabolic health.

## Figures and Tables

**Figure 1 nutrients-13-01748-f001:**
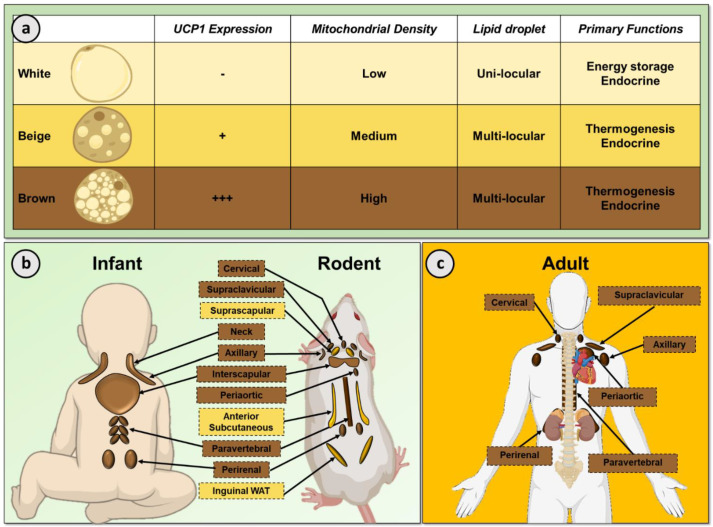
Adipose tissue function and location. (**a**) Brown adipocytes contain many UCP1 positive mitochondria, numerous small lipid droplets and are involved in thermogenesis. In contrast, white adipocytes contain few mitochondria that do not express UCP1 and a single, large lipid droplet for triglyceride storage. Beige adipocytes have an intermediate phenotype between classic brown and white adipocytes. (**b**) BAT is primarily located in the interscapular, neck, axillary and perirenal regions (brown boxes) in the human infant, with smaller depots behind the sternum and along the spine (human newborn) [7,8,9]. The size and composition of rodent BAT differs with age, sex and strain with the largest depot in the interscapular region (classical BAT) and smaller depots in the cervical, supraclavicular and peri-aortic regions. Inducible thermogenic beige adipocytes can form in specific WAT depots such as the suprascapular, anterior subcutaneous and inguinal regions (yellow boxes) [10,11]. (**c**) Adult human BAT is mainly located in the cervical, axillary and supraclavicular regions, with smaller depots observed in the periaortic, perirenal and paravertebral regions. Uncoupling protein 1 (UCP1), brown adipose tissue (BAT), white adipose tissue (WAT).

**Figure 2 nutrients-13-01748-f002:**
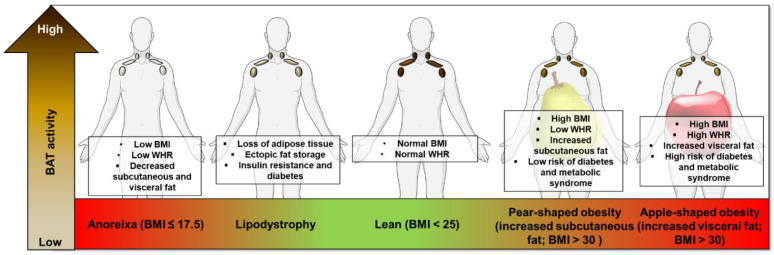
Summary of the impact of undernutrition, overnutrition and altered fat distribution on BAT function. BAT activity is decreased in disorders of both reduced and excessive WAT mass compared with normal weight individuals. Fat distribution may also regulate BAT function, for example central (apple-shaped) obesity may impair BAT activity to a greater extent than those with predominant peripheral (pear-shaped) obesity. In addition, lipodystrophy syndromes are associated with reduced BAT function. Brown adipose tissue (BAT), white adipose tissue (WAT), Body mass index (BMI), waist-hip-ratio (WHR).

**Figure 3 nutrients-13-01748-f003:**
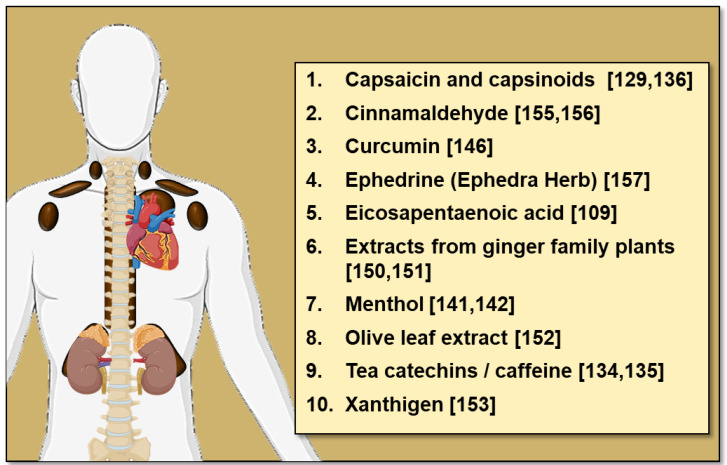
Dietary compounds with thermogenic potential in humans.

## Data Availability

Not applicable.

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
