# Peer review of "Nutritional Regulation of Human Brown Adipose Tissue"

_nutrients, 2021, doi:10.3390/nu13061748_

Round 1

Reviewer 1 Report

This is an excellent, well written and comprehensive review on the nutritional regulation of brown adipose tissue including an overview of how macronutrients and bioactive compounds regulate BAT. For once, I have no comments to make, this was a pleasure to read.

Reviewer 2 Report

This is a timely review of nutritional and exercise impacts on brown adipose tissue thermogenesis. As it is, the review is a little superficial but can be improved through the following: i. Greater critical review of the literature including description of key studies and their outcomes e.g. how was cold exposure used, are there any randomised controlled trials ; ii. Greater description of dose-response relationships e.g. how much cold exposure, what dose of capsaicin, curcumin etc have ben used in human studies; iii. Highlighting potential safety issues of different macro and micronutrient approaches that have been used; iv. Evidence of ethnic differences in BAT and responses to various measures to alter BAT function.

Reviewer 3 Report

This review seeks to provide an overview of nutritional influences on brown adipose tissue activity.  Topics covered include pathological conditions of nutritional excess (obesity), undernutrition (anorexia), and altered fat physiology (lipodystrophy) as well as impacts of dietary composition (macronutrients and bioactive compounds).  An overarching focus is directed toward identifying potential nutritional (non-pharmaceutical) approaches to enhance BAT activation for improved metabolic health through activation of BAT.  Human data is highlighted but reference to effects demonstrated in animal models is also included.  Several suggestions for improvement are detailed below.

Major

  1. Section 3, this section describes data showing an association between BAT FDG uptake and WAT distribution, an association of UCP1 expression in BAT with fat distribution (greater central/peripheral distribution) and a correlation of central obesity with cardiometabolic risk. From these observations it is suggested that altered BAT function is implicated in increased cardiometabolic risk.  On the other hand, Figure 2 suggests that there is an “impact of undernutrition, overnutrition and altered fat distribution on BAT function”.   This is a bit confusing, is the suggestion that BAT regulation is driving the altered fat distribution or that the fat distribution is driving altered BAT regulation?  Given the purely correlational data it is not possible to determine, if either possibility is the case or whether they are simply correlated but not causally connected.  Additionally, it is unclear how BAT function would be mechanistically connected to cardiometabolic risk; this should be more explicity stated.
  2. Line 170-172. The suggestion that caloric restriction may increase activation of BAT in obese subjects needs to include a reference supporting it or be more clearly stated as a speculative statement.  The data supporting an effect of CR to increase activation of BAT is very sparse, this assertion is largely based on one study in rats showing that the weight, but not UCP1 levels, of BAT is increased by caloric restriction.
  3. The section on dietary components affecting BAT activity is an important component of this review. This section might benefit from including more references to the rodent literature.  For example, several groups have shown that insulin can increase BAT activity (Holt & York, Br Res, 1989; Sakaguchi & Bray, JCI 1998; Rahmouni et al JCI 2004).  These examples provide a better index of BAT activity compared to FDG uptake, since effects of insulin to increase FDG uptake in BAT without concurrent activation of the tissue could confound interpretation of the effectiveness of insulin to activate BAT.  Additionally, since increases in UCP1 mRNA are only an index of BAT activation and can be dissociated from UCP protein levels (see Sivitz et al Endocrinology 1999) it is important to reference data showing direct measurements of BAT thermogenesis.  For example, Blouet & Schwartz (PLOS one, 2012) have shown directly that dietary fat induces activation of BAT thermogenesis via CCK release.
  4. It may be important to note that acute effects of diet-induced thermogenesis may differ from chronic maintenance on diets of differing macronutrient content. Perhaps similar to acute (BAT activating) vs. chronic (BAT inhibiting) effects of glucocorticoids. Along these lines, several groups have demonstrated that, similar to obese humans, rats chronically maintained on a high fat diet have impaired activation of brown adipose tissue (Levin et al, AJP, 1985; Sakaguchi et al Physiol Behav, 1989; Madden & Morrison, AJP 2016; Conceicao et al, eNeuro, 2021).
  5. The point of section “4.5 Protein” is not clear. On the one hand it seems to suggest that BAT activity is increased with low protein diet but on the other hand that increasing dietary protein might increase BAT activity.  Citation of the animal literature here might provide insight (for example: see, Glick & Raum, AJP, 1986).  Also, since protein would elicit CCK release and CCK activates BAT thermogenesis (Wang et al JCI Insight, 2019), it seems likely that protein consumption would increase BAT thermogenesis.
  6. Capsaicin has been suggested to increase BAT activation, however there is also data to suggest opposite effects of activation of TRPV1 (see, Mohammed et al AJP, 2018; Steiner et al, J Neurosci, 2007).

Minor

  1. Figure 1 typo in panel C “Axiallary”
  2. References 135-140 are not in the text, only in Figure 3.

Round 2

Reviewer 3 Report

All previous concerns have been adequately addressed.

Author Response

Many thanks for taking the time to review our manuscript. We are pleased that all previous concerns have been adequately addressed. 
